# Spatial distribution and determinants of unimproved sanitation facilities among households in Somalia: Using Somalia integrated household budget survey (SIHBS 2022)

Omar Muhumed Maidhane[1,2]*, Omran Salih[1,3], Abdisalam Hassan Muse[1,4], Abdirahman Omer Osman[1], Muse H. Abdi[1], Mahdi Hashi Hassan[1], Nur Mohamud Ali[1], Shacban Abdilahi Elmi[1]

1 School of Postgraduate Studies and Research, Amoud University, Borama, Somalia, 2 Research and Innovation Center, Galkayo University, Galmudug, Somalia, 3 Institute of System Science, Durban University of Technology, Durban, South Africa, 4 Research and Innovation Center, Amoud University, Borama, Somalia

* maidhane112@gmail.com

## Abstract

### Background

Access to adequate sanitation remains a critical public health challenge in Somalia, where a large portion of the population relies on unimproved facilities due to persistent conflict, climate shocks, and political instability. This reliance contributes to a high burden of waterborne diseases. This study aimed to assess the spatial distribution of unimproved sanitation and identify its individual and community-level determinants using recent national data to inform targeted interventions.

### Methods

This study is a secondary analysis of the 2022 Somalia Integrated Household Budget Survey (SIHBS), which included 7,212 households. The primary outcome was the use of unimproved sanitation facilities, categorized according to the WHO/UNICEF Joint Monitoring Programme (JMP) definitions. We employed a multilevel logistic regression model to identify individual and community-level determinants associated with unimproved sanitation. To analyze the spatial patterns of unimproved sanitation, we used Global Moran's I for spatial autocorrelation and the Getis-Ord Gi* statistic for hotspot analysis.

### Results

Overall, 36.87% of Somali households use unimproved sanitation facilities. There are significant disparities across residence types, with the highest prevalence among nomadic populations (83.28%), followed by rural (51.10%) and urban (23.88%)

**Data availability statement:** The data underlying the results presented in the study are available from the Somali National Bureau of Statistics (SNBS) [8,10,16]. The dataset, the Somalia Integrated Household Budget Survey (SIHBS) 2022, can be requested by researchers through the Somali National Data Archive (SoNADA) portal on the SNBS website [20].

**Funding:** The author(s) received no specific funding for this work.

**Competing interests:** The authors have declared that no competing interests.

residents. The multilevel analysis revealed that households in permanent/formal housing (AOR: 3.42) and those with IDP status (AOR: 3.18) had significantly higher odds of using unimproved sanitation. At the community level, urban residence was paradoxically associated with higher odds of unimproved sanitation (AOR: 7.99) compared to rural areas, while nomadic populations had significantly lower odds (AOR: 0.04), likely reflecting a high prevalence of open defecation not captured as a "facility." Spatial analysis identified significant hotspots of unimproved sanitation in the Hiraan (90.65%) and Bay (80.39%) regions, and cold spots in Banadir (5.37%) and Lower Shabelle (3.70%).

## Conclusion

The findings highlight deep inequalities in sanitation access across Somalia, driven by geographic location, socioeconomic status, and population group. The high prevalence of unimproved sanitation, especially among nomadic, rural, and displaced populations, calls for urgent, geographically-targeted interventions. A multi-pronged approach is necessary, focusing on the specific needs of different communities and addressing the underlying structural and individual-level drivers of poor sanitation to advance public health and sustainable development goals in the region.

---

## 1. Introduction

Access to adequate and equitable sanitation is a fundamental human right and a critical component of public health and sustainable development [1]. Globally, 3.5 billion people still lacked safely managed sanitation in 2022, with 419 million practicing open defecation [2]. The disease burden resulting from inadequate sanitation is immense, contributing to the transmission of diseases like cholera, typhoid, and intestinal worm infections, which disproportionately affect children under five [3,4]. Poor sanitation is a significant factor in environmental degradation and is closely linked to childhood stunting and malnutrition [8,17]. The Sustainable Development Goals (SDGs), particularly Target 6.2, aim to achieve universal access to sanitation for all and end open defecation by 2030, paying special attention to the needs of women, girls, and those in vulnerable situations [5].

Sub-Saharan Africa faces the most significant challenges in sanitation coverage, with disparities between urban and rural areas and slow overall progress [6,19]. Somalia, in particular, confronts acute sanitation challenges exacerbated by decades of conflict, political instability, climate-related disasters such as recurrent droughts, and large-scale displacement [1,22]. According to the WHO/UNICEF Joint Monitoring Programme (JMP), access to even basic sanitation facilities in Somalia is critically low, with a significant portion of the population, especially in rural and nomadic communities, resorting to unimproved facilities or open defecation [7,13]. Recent data from the 2020 Somalia Demographic and Health Survey indicated that a substantial proportion (59.7%) of households utilized unimproved sanitation facilities [8]. This

situation poses severe risks to public health, dignity, and safety, particularly for women and girls who face heightened risks of harassment and violence when seeking private places to relieve themselves [9].

Despite the clear urgency, there is limited comprehensive research on the specific factors driving the prevalence of unimproved sanitation in Somalia. While several studies have highlighted the scale of the problem, a detailed analysis that combines both spatial and multilevel approaches to understand the geographic disparities and the interplay of individual and community-level determinants is lacking [10,20]. Such an analysis is crucial for designing evidence-based, targeted interventions that can address the root causes of poor sanitation. Therefore, this study aims to assess the spatial distribution and identify the determinants of unimproved sanitation facilities among households in Somalia, using recent data from the Somalia Integrated Household Budget Survey (SIHBS 2022).

This study provides a novel and in-depth analysis of sanitation challenges in Somalia by being one of the first to combine a multilevel modeling approach with advanced spatial analysis. While previous research has acknowledged the poor state of sanitation in the country, this study moves beyond descriptive statistics to simultaneously investigate both individual and community-level determinants of unimproved sanitation. The use of the recent and comprehensive 2022 Somalia Integrated Household Budget Survey (SIHBS) dataset allows for a more current and granular understanding of the issue. Furthermore, the application of geospatial techniques such as Getis-Ord Gi* hotspot analysis and spatial autocorrelation provides a clear visual and statistical identification of geographic disparities, pinpointing specific regions and communities that are most affected. This dual approach offers a more robust evidence base for policymakers and public health stakeholders, enabling the design of targeted, context-specific interventions that address both the socioeconomic drivers and the geographical clustering of poor sanitation in Somalia.

## 2. Materials and methods

### 2.1 Study area

The study was conducted in Somalia, a country in the Horn of Africa with an estimated 2024 population of around 19 million and a low population density of approximately 32 people per square kilometer. The nation's economy is primarily agrarian, with the livestock sector accounting for a significant portion of the GDP and over half of export earnings. Despite recent economic growth, Somalia is classified as a least-developed country, and in 2022, over half of the population lived below the national poverty line. Decades of civil conflict and severe climate shocks have devastated public infrastructure, including water and sanitation systems, leading to widespread displacement and vulnerability among its diverse urban, rural, and nomadic communities [5].

### 2.2 Study design and data source

This study employed a secondary analysis of data from the 2022 Somalia Integrated Household Budget Survey (SIHBS), a nationally representative, community-based cross-sectional survey. The SIHBS was conducted by the Somalia National Bureau of Statistics (SNBS) and is the first comprehensive household budget survey in the country since 1985 [11].

The survey used a two-stage stratified cluster sampling design, covering urban, rural, and nomadic populations. In the first stage, enumeration areas (EAs) were selected with probability proportional to their size, followed by the systematic selection of a fixed number of households from each EA in the second stage. The survey included a total sample of 7,212 households. This analysis utilized the final weighted sample to ensure national representativeness.

### 2.3 Data validation

To ensure the findings are representative of the national population and to account for the complex survey design, sampling weights provided in the SIHBS dataset were applied during all statistical analyses. This weighting procedure adjusts for the unequal probability of selection and non-response rates, ensuring reliable and valid estimates.

## 2.4 Study variables

The primary outcome variable was the status of the household sanitation facility, categorized as a binary variable: "unimproved" or "improved." This classification was based on the WHO/ UNICEF JMP definitions [12]. "Unimproved sanitation facilities" were defined as pit latrines without a slab or platform, hanging latrines, bucket latrines, and the practice of open defecation [13]. *This was coded as '1'.* "Improved sanitation facilities" included flush or pour-flush toilets connected to a piped sewer system, septic tanks, or pit latrines; ventilated improved pit latrines; and pit latrines with a slab, which are not shared with other households. *This was coded as '0'.*

The independent variables were classified into individual-level and community-level factors. Individual-level variables included the sex of the household head (male or female), age of the household head (categorized), educational attainment (no education, primary, secondary, or higher), household size, and the household wealth index, which was categorized into five quintiles: poorest, poorer, middle, richer, and richest. Community-level variables comprised the place of residence (urban, rural, or nomadic), region or federal member state, and community media exposure, defined as the proportion of households within the enumeration area (EA) that had access to radio or television.

## 2.5 Data analysis

### 2.5.1 Univariant and multilevel analysis.
Data cleaning, coding, and analysis were performed using STATA version 17. Descriptive statistics, including frequencies and percentages, were used to summarize the characteristics of the study population and the prevalence of unimproved sanitation. Due to the hierarchical nature of the survey data, where households (level 1) are nested within enumeration areas or clusters (level 2), a multilevel logistic regression model was employed to identify the determinants of unimproved sanitation [14]. This approach accounts for the clustering effect and avoids the underestimation of standard errors that can occur with standard logistic regression [8,12].

Four consecutive models were fitted: a null model (Model 0) with no explanatory variables to assess the variance in unimproved sanitation between clusters; Model I included only individual-level variables; Model II included only community-level variables; and Model III, the final model, included both individual and community-level variables simultaneously. Measures of variation (random effects) such as the Intraclass Correlation Coefficient (ICC), Median Odds Ratio (MOR), and Proportional Change in Variance (PCV) were calculated to quantify the extent of clustering. Model fit was assessed using the Akaike information criterion (AIC) and Bayesian Information criterion (BIC), with the model having the lowest AIC and BIC values considered the best fit. Adjusted Odds Ratios (AOR) with 95% confidence intervals (CI) were used to report the final associations, and a p-value of < 0.05 was considered statistically significant.

### 2.5.2 Spatial analysis.
Geospatial analysis was conducted to explore the spatial distribution and identify geographic hotspots of unimproved sanitation facilities in Somalia using ArcGIS version 10.7 and SaTScan version 10.1. First, spatial autocorrelation was assessed using the Global Moran's I statistic to determine if the pattern of unimproved sanitation was clustered, dispersed, or random. A statistically significant and positive Moran's I value indicates spatial clustering of similar values [15].

Second, Getis-Ord Gi* hotspot analysis was performed to identify the locations of statistically significant hot spots (areas with high prevalence of unimproved sanitation) and cold spots (areas with low prevalence) [15]. Third, ordinary Kriging spatial interpolation was used to predict the prevalence of unimproved sanitation in unsampled areas based on the values from sampled clusters, creating a continuous risk map for the entire country. Finally, a purely spatial scan statistic using a Bernoulli model was employed in SaTScan software to detect significant spatial clusters of unimproved sanitation. Households with unimproved facilities were treated as cases, and those with improved facilities as controls, to identify geographic areas with a higher-than-expected prevalence.

## 2.6 Ethical statement

This study is based on a secondary analysis of publicly available and anonymized data from the Somalia Integrated Household Budget Survey (SIHBS 2022). The original survey protocol was reviewed and approved by the relevant institutional review board in Somalia. The implementing agency, the Somalia National Bureau of Statistics (SNBS), obtained informed consent from all participants before data collection. Permission to use the dataset for this study was obtained through the official data archive channels.

## 3. Results

### 3.1 Univariate analysis

The univariate analysis, detailed in Table 1, reveals that both individual and community-level characteristics are significantly associated with the use of unimproved sanitation facilities in Somalia. All variables presented demonstrated a statistically significant relationship with the outcome ($p < 0.001$ for most, with minor exceptions), underscoring the multifaceted nature of sanitation access.

Overall, 36.87% of Somali households rely on unimproved sanitation. This prevalence is situated within a range of recent estimates, being higher than the 22.3% reported in an analysis of the 2020 Somalia Health and Demographic Survey (SHDS) data [16] but lower than the 59.7% from the same 2020 survey data reported by another study [8]. These discrepancies may reflect differences in variable definitions, weighting methodologies, or the dynamic nature of the country's humanitarian landscape between 2020 and the 2022 SIHBS survey period.

At the individual level, socioeconomic indicators emerged as powerful determinants. Education was a critical factor, with households headed by an individual with no formal schooling being significantly more likely to use unimproved facilities (44.02%) compared to those with some education (30.12%). This finding aligns with extensive research across sub-Saharan Africa, which consistently shows that education enhances awareness of hygiene-related health risks and often correlates with higher economic status, enabling investment in better facilities [3,8]. Similarly, proxies for wealth, such as access to electricity and internet use, were strongly correlated with improved sanitation. For instance, nearly 60% of households without electricity used unimproved facilities, compared to just 23.12% of those with access. This stark divide highlights that unimproved sanitation is deeply embedded within a broader context of poverty and lack of access to modern infrastructure.

Vulnerability was another key theme. The data starkly illustrates the plight of Internally Displaced Persons (IDPs), with 64.27% of IDP households using unimproved facilities, compared to 35.64% of non-IDP households. This corroborates findings from humanitarian reports and studies on Somalia, which document the dire sanitation conditions in displacement camps where basic services are often overwhelmed or absent [16]. Furthermore, food-insecure households reported significantly higher rates of unimproved sanitation (46.76%) than food-secure households (25.17%), underscoring a syndemic relationship where different forms of deprivation—lack of food, shelter, and sanitation—cluster within the same vulnerable populations. Housing type also reflected this, with over 61% of those in temporary or informal housing lacking improved sanitation, a finding consistent with studies on informal settlements globally [17].

At the community level, the type of residence was the most dramatic determinant. Nomadic populations face the most extreme challenges, with an overwhelming 83.28% using unimproved sanitation. This is consistent with studies on pastoralist communities, whose mobile lifestyle makes investment in permanent sanitation infrastructure impractical, often leading to a high prevalence of open defecation [3]. A significant rural-urban divide was also evident, with 51.10% of rural households using unimproved facilities compared to 23.88% of urban households. This rural disparity is a well-documented phenomenon across sub-Saharan Africa, driven by lower investment, weaker governance, and less access to markets for sanitation products and services [13,17].

 

**Table 1. Descriptive Statistics of Unimproved Sanitation Facilities in Somalia.**

| Variable *Individual-level variables* | Categories | Sanitation Facilities | | Chi2 | P-value |
|---|---|---|---|---|---|
| | | Unimproved (%) | Improved (%) | | |
| Sanitation Facility | | 36.87 | 63.13 | | |
| Age of household head | <20 years | 32.74 | 67.26 | 47.3653 | 0.001 |
| | 20-29 years | 34.71 | 65.29 | | |
| | 30-39 years | 38.97 | 61.03 | | |
| | 40-49 years | 40.45 | 59.55 | | |
| | 50-59 years | 41.59 | 58.41 | | |
| | 60+years | 37.64 | 62.36 | | |
| Sex of household head | Male | 38.58 | 61.42 | 12.7784 | 0.001 |
| | Female | 35.42 | 64.58 | | |
| Ever Attended school | Yes | 30.12 | 69.88 | 247.5964 | 0.001 |
| | No | 44.02 | 55.98 | | |
| Marital Status | Married | 42.66 | 57.34 | 153.5151 | 0.001 |
| | Divorced | 31.27 | 68.73 | | |
| | Never married | 31.51 | 68.49 | | |
| | Widowed | 33.58 | 66.42 | | |
| Housing type category | Temporary/Informal/Basic | 61.32 | 38.68 | 1300 | 0.001 |
| | Permanent/Formal | 26.79 | 73.21 | | |
| Lighting-source | Traditional/Other | 58.86 | 41.14 | 1200 | 0.001 |
| | Modern/Improved | 26.15 | 73.85 | | |
| Household Ownership | Yes | 36.53 | 63.47 | 4.7038 | 0.030 |
| | No,non-household member | 39.56 | 60.44 | | |
| IDP | Yes | 64.27 | 35.73 | 173.5672 | 0.001 |
| | No | 35.64 | 64.36 | | |
| Use Internet | Yes | 21.62 | 78.38 | 691.7784 | 0.001 |
| | No | 45.71 | 54.29 | | |
| Electricity Access | Yes | 23.12 | 76.88 | 1600 | 0.001 |
| | No | 59.58 | 40.42 | | |
| Labour_status | Yes | 39.04 | 60.96 | 3.6606 | 0.056 |
| | No | 36.54 | 63.46 | | |
| Food security | Food Secure | 25.17 | 74.83 | 593.9134 | 0.001 |
| | Food Insecure | 46.76 | 53.24 | | |
| *Community Level Variables* | | | | | |
| Type of Residence | Rural | 51.10 | 48.90 | 1800 | 0.001 |
| | Urban | 23.88 | 76.12 | | |
| | Nomadic | 83.28 | 16.72 | | |
| Region | Awdal | 44.13 | 55.87 | 2600 | 0.001 |
| | Bakool | 42.61 | 57.39 | | |
| | Banadir | 5.37 | 94.63 | | |
| | Bari | 11.38 | 88.62 | | |
| | Bay | 80.39 | 19.61 | | |
| | Galgaduud | 36.57 | 63.43 | | |
| | Gedo | 21.27 | 78.73 | | |
| | Hiraan | 90.65 | 9.35 | | |
| | Lower Juba | 53.09 | 46.91 | | |
| | Lower Shabelle | 3.70 | 96.30 | | |

*(Continued)*

**Table 1.** (Continued)

| Variable<br>*Individual-level variables* | Categories | Sanitation Facilities | | Chi2 | P-value |
|---|---|---|---|---|---|
| | | Unimproved (%) | Improved (%) | | |
| | Waqooyi Galbeed | 41.67 | 58.33 | | |
| | Middle Shabelle | 18.68 | 81.32 | | |
| | Mudug | 29.01 | 70.99 | | |
| | Nugaal | 52.02 | 47.98 | | |
| | Sanaag | 50.54 | 49.46 | | |
| | Sool | 27.62 | 72.38 | | |
| | Togdheer | 27.78 | 72.22 | | |

Finally, the analysis revealed profound regional inequalities. The prevalence of unimproved sanitation ranged from an alarming 90.65% in Hiraan and 80.39% in Bay to as low as 3.70% in Lower Shabelle and 5.37% in the capital region of Banadir. These geographical disparities are likely a direct reflection of varying levels of stability, governance, conflict, and humanitarian presence across Somalia's federal member states. Regions like Hiraan and Bay have been epicenters of conflict and climate-related shocks, severely hampering infrastructure development, whereas more stable and urbanized regions like Banadir have benefited from greater investment and access to services. This wide variation underscores that a single national figure for sanitation access masks critical local realities and highlights the necessity for geographically-targeted interventions to address the diverse challenges across Somalia.

### 3.2 Multilevel logistic regression

The multilevel logistic regression analysis was conducted in a stepwise manner to disentangle the individual and community-level determinants of unimproved sanitation in Somalia, with the results presented in Table 2. This approach allows for a comparison between models to determine the most reliable and comprehensive explanation for the observed sanitation disparities.The process began with an empty model (Model 0), which contained no predictor variables. This model revealed a very high intraclass correlation coefficient (ICC) of 0.94, indicating that 94% of the total variance in the use of unimproved sanitation can be attributed to differences between communities (enumeration areas). This finding strongly justified the use of a multilevel model, as it confirms that household sanitation status is heavily dependent on community-level context.

**Model 1** focused exclusively on individual-level factors. This model identified households living in permanent/formal housing (AOR: 3.67; 95% CI: 2.97, 4.53) and those with internally displaced person (IDP) status (AOR: 3.37; 95% CI: 2.16, 5.23) as having significantly higher odds of using unimproved sanitation. This highlights the severe vulnerability of displaced populations and suggests a paradox where "permanent" housing in camp-like settings may lack basic services. Conversely, factors often associated with poverty, such as lacking electricity (AOR: 0.39) or internet access (AOR: 0.68), were paradoxically associated with lower odds, likely because these households are concentrated in areas with different sanitation challenges (e.g., rural open defecation) not fully captured by this model's scope. While insightful, Model 1 only tells part of the story by ignoring the broader community context.

The results from **Model 2**, which incorporates community-level factors, indicate that contextual characteristics play a substantial role in shaping the outcome, with several variables demonstrating strong and statistically significant associations. Notably, type of residence emerged as a key determinant, with households in urban areas exhibiting significantly higher odds (AOR = 14.46; 95% CI: 7.70–27.15) compared to the reference category, while nomadic populations showed markedly lower odds (AOR = 0.01; 95% CI: 0.004–0.035), highlighting stark disparities across settlement types. Regional variations were also pronounced, with exceptionally high odds observed in regions such as Lower Shabelle

**Table 2. Multilevel Analysis Unimproved Sanitation Facilities in Somalia.**

| Variable | Categories | Model 0: Empty Model (AOR [95% CL]) | Model 1: Individual Factors (AOR [95% CL]) | Model 2: Community Factors (AOR [95% CL]) | Model 3: Full Model (AOR [95% CL]) |
|---|---|---|---|---|---|
| Age of household head | 20-29 years | | 0.83 [0.67, 1.04] | | 0.84 [0.67,1.05] |
| | 30-39 years | | 0.82 [0.64, 1.06] | | 0.81 [0.63,1.04] |
| | 40-49 years | | 0.94 [0.69, 1.27] | | 0.92 [0.68,1.25] |
| | 50-59 years | | 0.83 [0.60, 1.16] | | 0.81 [0.58,1.13] |
| | 60 + years | | 0.89 [0.65, 1.20] | | 0.86 [0.63, 1.16] |
| Sex of household head | Female | | 1.14 [0.99, 1.31] | | 1.13 [0.98, 1.30] |
| Ever Attended school | No | | 1.03 [0.87, 1.21] | | 1.02 [0.87, 1.20] |
| Marital Status | Divorced | | 0.90 [0.66, 1.23] | | 0.88 [0.64, 1.21] |
| | Never married | | 0.88 [0.70, 1.09] | | 0.85 [0.68, 1.06] |
| | Widowed | | 1.08 [0.82, 1.43] | | 1.06 [0.80, 1.40] |
| Housing type category | Permanent/Formal | | 3.67 [2.97, 4.53] | | 3.42 [2.76, 4.23] |
| Lighting-source category | Modern/Improved | | 0.88 [0.65, 1.19] | | 0.84 [0.62, 1.13] |
| Household Ownership | No, non-household member | | 0.69 [0.55, 0.86] | | 0.69 [0.55, 0.87] |
| IDP | No | | 3.37 [2.16, 5.23] | | 3.18 [2.06, 4.92] |
| Use Internet | No | | 0.68 [0.57, 0.80] | | 0.71 [0.60, 0.84] |
| Electricity Access | No | | 0.39 [0.28, 0.54] | | 0.44 [0.32, 0.60] |
| Labour_status | No | | 1.04 [0.85, 1.28] | | 1.04 [0.85, 1.28] |
| Food security | Food Insecure | | 0.63 [0.54, 0.75] | | 0.66 [0.56, 0.78] |
| Type of Residence | Urban | | | 14.46 [7.70, 27.15] | 7.99 [4.35,14.69] |
| | Nomadic | | | 0.01 [0.004, 0.035] | 0.04 [0.01,0.11] |
| Region | Bakool | | | 0.66 [0.14, 3.05] | 1.80 [0.41, 7.83] |
| | Banadir | | | 23.18 [5.05,106.45] | 17.12[3.88,75.61] |
| | Bari | | | 12.37 [2.67, 57.34] | 11.64[2.6, 52.6] |
| | Bay | | | 0.04 [0.01, 0.2] | 0.08[0.02, 0.39] |
| | Galgaduud | | | 1.73 [0.40, 7.44] | 2.39 [0.58, 9.82] |
| | Gedo | | | 6.01 [1.31, 27.46] | 11.98 [2.68,53.5] |
| | Hiraan | | | 0.0009 [0.0001, 0.0051] | 0.003[0.001,0.02] |
| | Lower Juba | | | 0.29 [0.06,1.46] | 0.37 [0.08, 1.70] |
| | Lower Shabelle | | | 213.19 [27.92,1627.65] | 175 [24.4,1259.9] |
| | Waqooyi Galbeed | | | 1.34 [0.35, 5.07] | 1.26 [0.35, 4.58] |
| | Middle Shabelle | | | 7.09 [1.63, 30.78] | 12.77 [3.02, 54] |
| | Mudug | | | 2.64 [0.57, 12.29] | 2.94 [0.66, 13.07] |
| | Nugaal | | | 0.15 [0.03, 0.73] | 0.15 [0.03, 0.67] |
| | Sanaag | | | 0.39 [0.08, 1.80] | 0.36 [0.08, 1.57] |
| | Sool | | | 4.96 [1.03, 23.78] | 4.75 [1.05, 21.41] |
| | Togdheer | | | 14.77 [3.21, 67.87] | 17.00 [3.80, 76.1] |

(AOR = 213.19; 95% CI: 27.92–1627.65), Banadir (AOR = 23.18; 95% CI: 5.05–106.45), and Bari (AOR = 12.37; 95% CI: 2.67–57.34), whereas regions like Bay and Hiraan demonstrated significantly reduced odds, suggesting substantial geographic inequalities. Overall, Model 2 underscores that beyond individual characteristics, structural and geographic factors—including place of residence, regional context are critical determinants, reflecting broader socio-economic and spatial inequalities within the study population.

**Model 3**, the full and most reliable model, integrated both individual and community-level variables. Its superiority is statistically confirmed by its significantly lower AIC (7356.637) and BIC (7637.396) values compared to all other models (as shown in Table 3), indicating the best model fit. By including both sets of factors, this model provides the most nuanced and accurate understanding of the issue.

Comparing **Model 1** and **Model 3** reveals both similarities and crucial differences in interpretation. The strong effects of individual-level factors like permanent/formal housing (AOR: 3.42) and IDP status (AOR: 3.18) remained highly significant and of a similar magnitude in Model 3. This similarity demonstrates that these household-level vulnerabilities are robust predictors of unimproved sanitation, even after accounting for community-level differences. Their influence is direct and powerful regardless of geographic location.

However, the inclusion of community-level variables in **Model 3** fundamentally changes the overall picture. The model shows that community context is an overwhelmingly powerful driver that reshapes the risk landscape. For instance, urban residence was associated with dramatically higher odds of unimproved sanitation (AOR: 7.99) compared to rural areas, while nomadic populations had exceptionally lower odds (AOR: 0.04), likely reflecting a high prevalence of open defecation not classified as a "facility." Most strikingly, regional disparities were extreme, with residents in Lower Shabelle having an astonishingly high odds (AOR: 175.42) compared to the reference region of Awdal, while those in Hiraan and Bay had significantly lower odds.

In conclusion, **Model 3 is the most reliable** because it accounts for the hierarchical nature of the data and demonstrates that while individual household characteristics are important, they operate within a much larger context of community and regional factors. The difference between Model 1 and Model 3 is that the latter clarifies that an individual's risk is profoundly modified by where they live. Therefore, a comprehensive understanding requires considering both the specific circumstances of a household (like displacement) and the overarching structural realities of their community (such as urban infrastructure deficits or regional instability).

### 3.3 Logistic regression models comparison

The model's comparison in Table 3 provides a clear progression in model fit and explanation of variance regarding unimproved sanitation facilities in Somalia. The initial Empty Model (Model 0), with a Log-likelihood of −3978.9811, an AIC of 7961.962, and a BIC of 7976.739, reveals a substantial intraclass correlation coefficient (ICC) of 0.9396. This high ICC indicates that approximately 94% of the total variance in unimproved sanitation is attributable to differences between clusters (enumeration areas), underscoring the necessity of a multilevel approach. As individual-level variables are introduced in Model 1, the Log-likelihood improves to −3797.8329, while the AIC (7635.666) and BIC (7783.434) decrease, signifying a better model fit. The ICC also drops to 0.8792, suggesting that individual household characteristics explain some of the variance in sanitation status. Model 2, incorporating only community-level variables, shows an even greater improvement, with a Log-likelihood of −3784.4915 and further reductions in AIC (7608.983) and BIC (7756.751). The ICC in this model is 0.7786, indicating that community-level factors account for a significant portion of the variability in unimproved sanitation. Finally, Model 3, the Full Model, which includes both individual and community-level variables, demonstrates the best fit among all models. It achieves the highest Log-likelihood of −3640.3183 and the lowest AIC (7356.637) and BIC (7637.396). The ICC in the Full Model is 0.7418, the lowest of all models, implying that the combined individual and community-level factors substantially reduce the

**Table 3. Models comparison.**

| Model | N | Log-likelihood | AIC | BIC | ICC |
|---|---|---|---|---|---|
| Model 0 (Empty) | 11,949 | −3978.9811 | 7961.962 | 7976.739 | 0.9396 |
| Model 1 (Individual) | 11,949 | −3797.8329 | 7635.666 | 7783.434 | 0.8792 |
| Model 2 (Community) | 11,949 | −3784.4915 | 7608.983 | 7756.751 | 0.7786 |
| Model 3 (Full) | 11,949 | −3640.3183 | 7356.637 | 7637.396 | 0.7418 |

unexplained variance between clusters. The consistent decrease in AIC, BIC, and ICC values across the models, culminating in the Full Model, confirms that a multilevel approach integrating both individual and community-level determinants is essential for comprehensively understanding the factors influencing unimproved sanitation facilities in Somalia.

### 3.4 Spatial analysis result

**3.4.1 Spatial distribution.** The spatial distribution of unimproved sanitation facilities in Somalia, depicted in Fig 1, reveals profound regional inequalities that mirror the country's complex landscape of security, governance, and development. The analysis identifies critical hotspots in the Hiraan (90.65%) and Bay (80.39%) regions, where access to basic sanitation is almost non-existent for the vast majority of the population. This dire situation is a direct consequence of these regions being epicenters of protracted conflict, severe climate shocks, and political instability. The persistent insecurity destroys existing infrastructure, prevents new development projects from being implemented, and severely limits the operational capacity of humanitarian and development organizations. Furthermore, these regions host large populations of internally displaced persons (IDPs), who are often congregated in camps where sanitation facilities are overwhelmed or entirely absent, thus driving up the regional prevalence of unimproved sanitation.

In stark contrast, the regions of Banadir (5.37%) and Lower Shabelle (3.70%) emerge as significant cold spots, indicating far better access to improved sanitation. The success in Banadir, which contains the capital city Mogadishu, can be

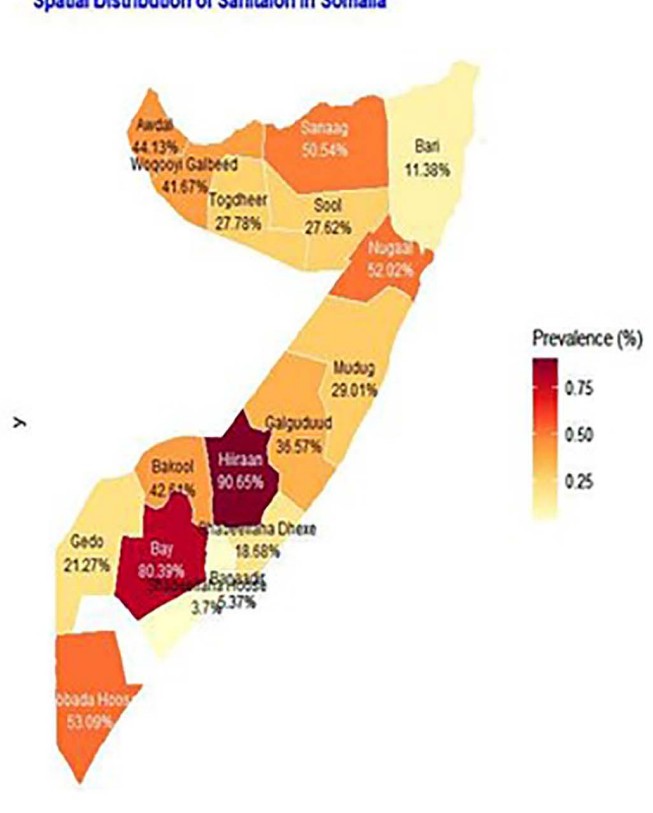

**Fig 1. Spatial Distribution of Unimproved Sanitation Facilities.**

attributed to its status as the seat of government and the primary hub for international partners, NGOs, and private invest-ment. This concentration of resources has led to greater investment in urban infrastructure and public services compared to other parts of the country. Similarly, the low prevalence in Lower Shabelle is likely influenced by its proximity to the capital and its role as a key agricultural region, which supports more settled communities and has historically seen greater development focus.

Between these extremes lie regions with moderate to high challenges, such as Lower Juba, Nugaal, and Sanaag, which all report prevalence rates above 50%. Ultimately, the map does not merely show sanitation coverage; it provides a clear visual proxy for underlying disparities in stability, governance, economic development, and humanitarian pres-ence across Somalia, underscoring the necessity of geographically-targeted interventions to address these deep-rooted inequalities.

**3.4.2 Spatial autocorrelation-global Moran's I.** Fig 2, the Global Moran's I normal distribution curve, indicates that the spatial distribution of unimproved sanitation facilities in Somalia is not statistically clustered or dispersed. With a Moran's Index of −0.19894, a z-score of −0.75544, and a high p-value of 0.77501, the data suggests a random spatial pattern, meaning there's no significant global clustering of high or low prevalence areas.

**3.4.3 Spatial autocorrelation-local Moran's I.** Fig 3 and Fig 4,composed of two sets of maps, illustrates the localized patterns of unimproved sanitation facilities in Somalia using Local Moran's I and corresponding p-values. The maps on the left side (grey with blue patches) identify specific areas where local spatial clustering is statistically significant, meaning the prevalence of unimproved sanitation in those blue regions is not random compared to their neighbors. The maps on the right, which are almost entirely dark blue, strongly indicate that for nearly all regions across Somalia, the observed local spatial patterns of unimproved sanitation are statistically significant (p-value less than 0.05). This means that, despite a lack of overall global clustering, there are distinct and statistically meaningful local concentrations either "hotspots" of high unimproved sanitation or "coldspots" of low unimproved sanitation across the vast majority of the country's administrative divisions.

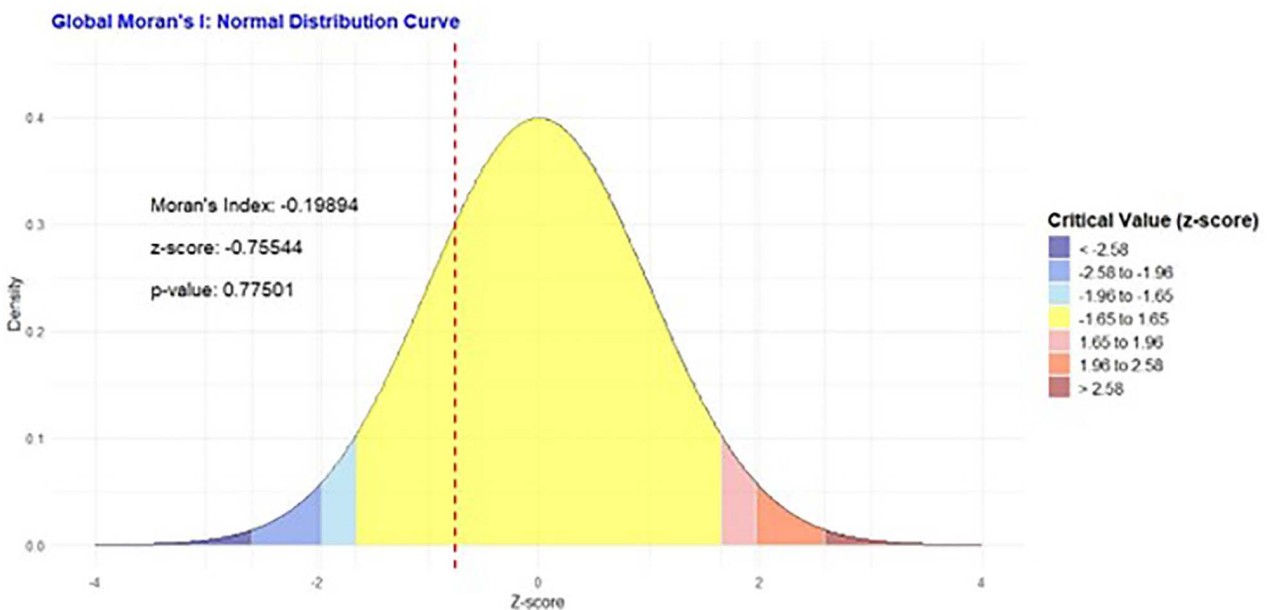

**Fig 2. Global Moran's I for spatial Autocorrelation of unimproved sanitation facilities.**

**Fig 3. Local Moran's I of unimproved sanitation.**

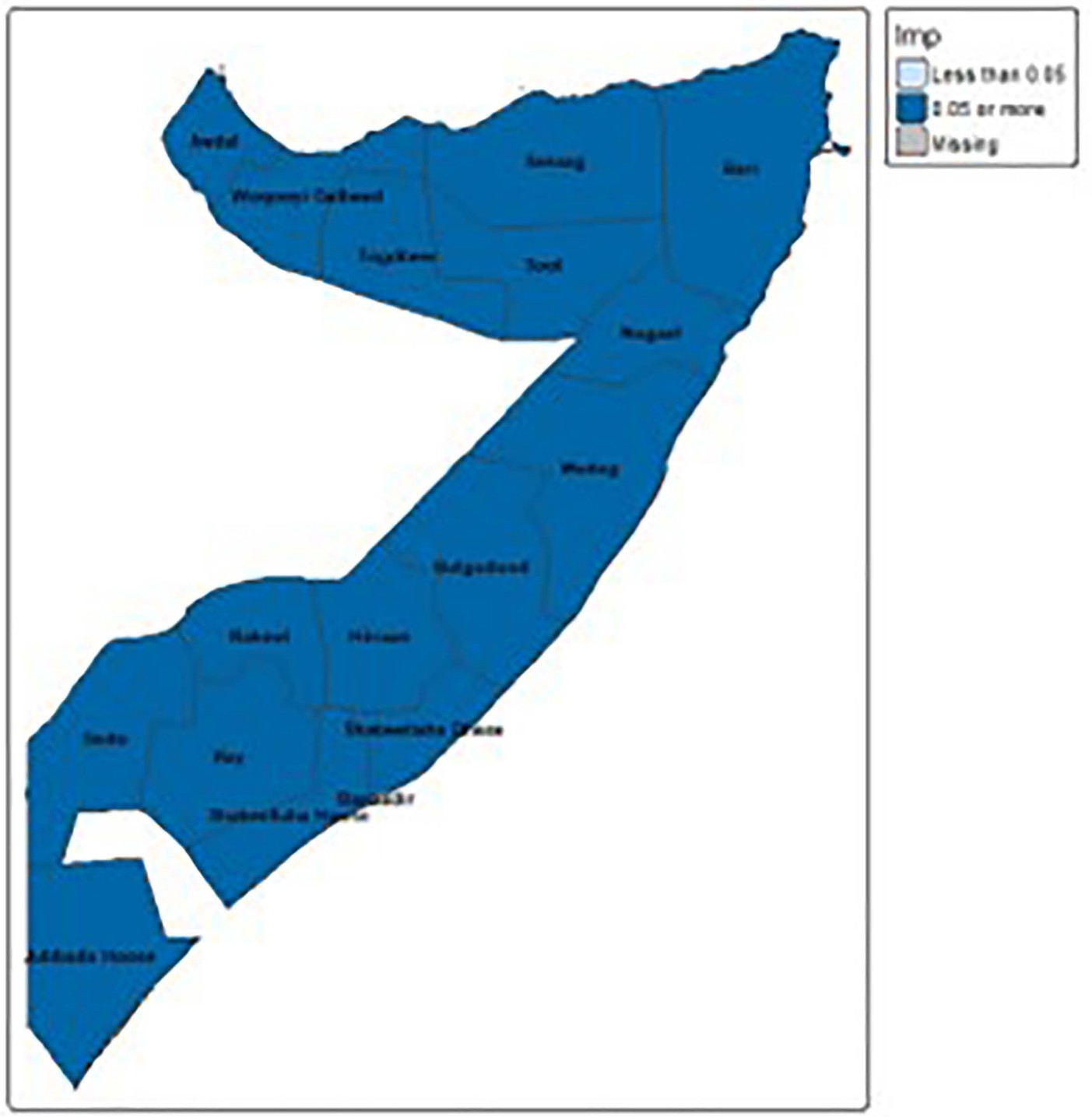

**Fig 4. Local Moran's I and its p-value for significance.**

**3.4.4 Hot spot and cold spot analysis.** Fig 5, displaying the Getis-Ord Gi* results, maps the spatial distribution of statistically significant hot spots and cold spots of unimproved sanitation facilities across Somalia. The color gradient in the legend indicates the range of Gi* values, which identify areas with high values (hot spots) or low values (cold spots) clustered together. Darker green shades (0.615 to 1.603) represent statistically significant hot spots, indicating regions where high prevalence of unimproved sanitation facilities is clustered. We can observe these hot spots prominently in areas like Hiraan, Bay, Galgaduud, Gedo, Mudug, and parts of the northern regions such as Nugaal. Conversely, the lighter purple shades (−1.847 to −0.979) represent statistically significant cold spots, signifying clusters of low prevalence

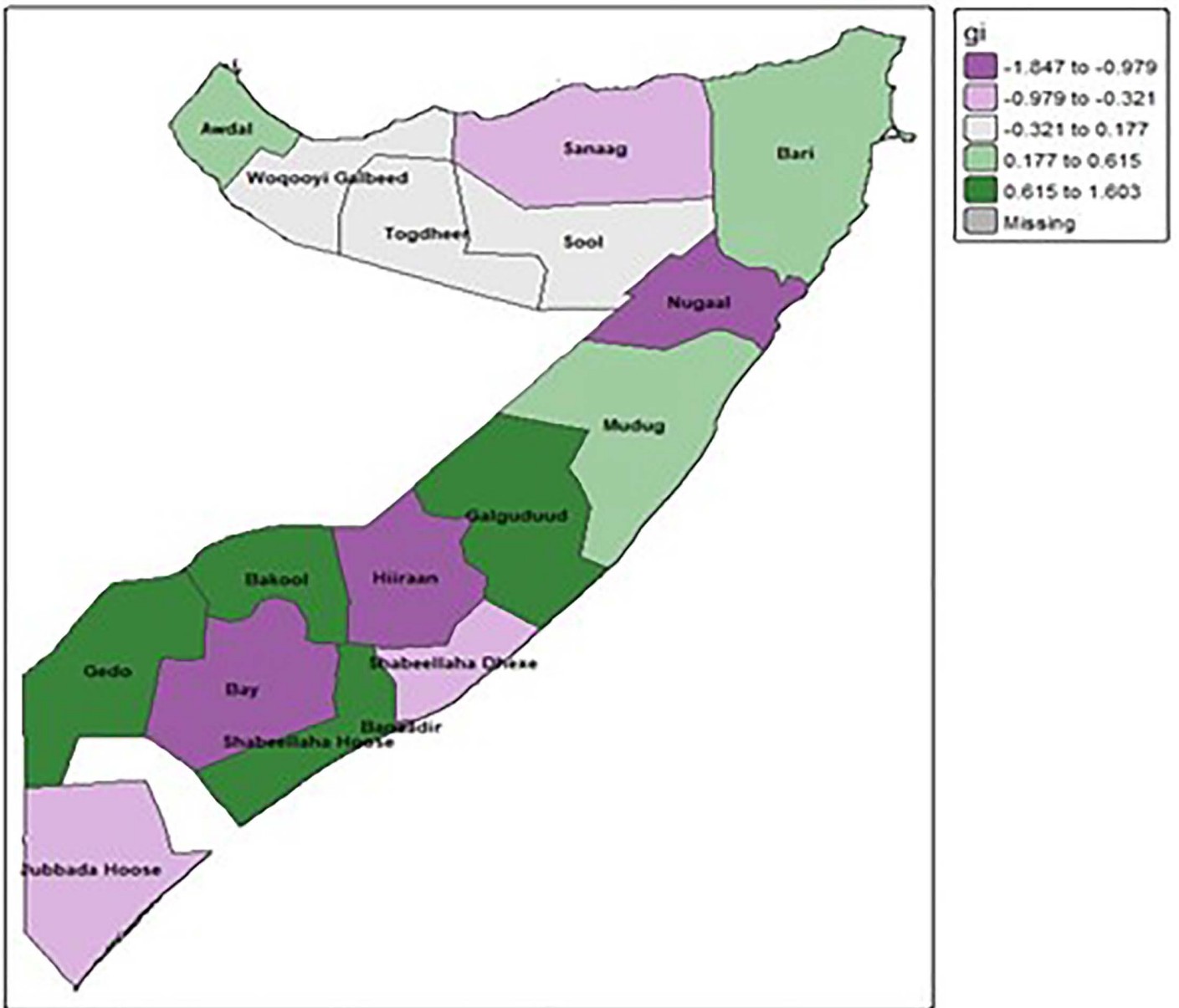

**Fig 5. Hot and cold spots region using Getis-Ord G\*.**

of unimproved sanitation facilities. These cold spots appear in regions like Awdal, Juba Hoose, and parts of the eastern coast (Bari). Intermediate shades of purple and green suggest areas with less extreme clustering or values closer to the mean. This Getis-Ord Gi* analysis visually corroborates and further refines the understanding of spatial disparities, clearly pinpointing specific regions that are hotbeds of unimproved sanitation and those with better access, thereby providing critical insights for geographically targeted public health interventions.

## 4. Discussion

This study's multilevel and spatial analysis provides a granular understanding of the complex determinants driving the use of unimproved sanitation in Somalia, with findings that both corroborate and challenge existing literature. The overall prevalence of over one-third of households using unimproved facilities highlights a persistent public health crisis. This figure is situated within a range of estimates for Somalia, including a 2024 study using 2020 data that reported 22.3% of households using unimproved facilities and an earlier survey indicating a higher rate of 59.7% [2,20]. These variations likely reflect differences in survey methodologies and the dynamic nature of displacement and infrastructure challenges in the country.

A key finding is the stark disparity across residence types, with nomadic populations exhibiting an overwhelming 83.28% prevalence of unimproved sanitation. This is consistent with the understanding that nomadic lifestyles present significant challenges for fixed sanitation infrastructure, often leading to a reliance on open defecation [3,17]. The significantly lower odds of unimproved sanitation among nomadic groups in the multilevel model (AOR: 0.04) is a statistical reflection of open defecation not being classified as a *facility*, rather than indicating better access. In contrast, the high odds for urban residents (AOR: 7.99) compared to their rural counterparts' points to a crisis of rapid, unplanned urbanization outstripping the provision of basic services, a phenomenon observed in other rapidly growing urban centers in sub-Saharan Africa. A recent study in Ethiopia, for example, found that while urban areas overall had better access, certain urban populations were highly vulnerable [6,18].

The multilevel analysis revealed several nuanced and seemingly paradoxical associations. The finding that households in permanent/formal housing had significantly higher odds of using unimproved sanitation (AOR: 3.42) is counterintuitive but may be explained by the context of protracted displacement. Many internally displaced persons (IDPs) live in camps where the structures are considered "permanent," yet they lack adequate sanitation services [19]. This is strongly supported by the finding that IDP status itself dramatically increases the odds of using unimproved sanitation by more than threefold (AOR: 3.18), confirming the extreme vulnerability of this population group, a fact well-documented in humanitarian reports on Somalia [7,20].

The spatial analysis was instrumental in moving beyond national averages to identify specific geographic inequities. The identification of Hiraan (90.65%) and Bay (80.39%) as critical hotspots aligns with the known impacts of conflict, climate shocks, and instability on these regions, which severely impede infrastructure development. Conversely, the significantly better outcomes in the more urbanized regions of Banadir (5.37%) and Lower Shabelle (3.70%) underscore the importance of governance and the presence of humanitarian and development actors. The Getis-Ord Gi* analysis, by confirming these statistically significant hot and cold spots, provides a clear evidence base for geographically targeted interventions. While the Global Moran's I indicated no overall global clustering, the significant local clustering reinforces the idea that the drivers of poor sanitation are highly localized. This aligns with broader research across sub-Saharan Africa, which increasingly calls for context-specific, decentralized approaches to address sanitation challenges effectively [21–23].

## 5. Conclusion

This study reveals that the state of sanitation in Somalia is a crisis of deep and geographically entrenched inequalities, driven by a complex interplay of socioeconomic vulnerability, displacement, and lifestyle. The overall prevalence of over

one-third of households using unimproved sanitation masks profound disparities, with the burden falling disproportionately on nomadic populations (83.28%), rural communities, and the internally displaced. The multilevel analysis moved beyond simple prevalence to uncover critical, and at times paradoxical, determinants. Counterintuitively, households in permanent/formal housing and those with IDP status faced significantly higher odds of using unimproved facilities. This highlights a critical reality in Somalia: many IDPs reside in camp-like settings with "permanent" structures that lack basic services, underscoring that housing without integrated sanitation infrastructure exacerbates vulnerability.

Furthermore, the findings challenge a simplistic urban-rural dichotomy. While urban areas have better sanitation coverage overall, the multilevel model showed urban residence was paradoxically associated with higher odds of unimproved sanitation compared to rural areas. This points to a crisis of rapid, unplanned urbanization where the growth of informal settlements is far outstripping the provision of essential services. Similarly, while nomadic populations have the highest prevalence of unimproved sanitation, their exceptionally low odds in the model reflect a statistical nuance where the widespread practice of open defecation is not captured as a "facility," rather than indicating better access.

The spatial analysis provides a clear roadmap for intervention, confirming that the drivers of poor sanitation are highly localized. While no single national pattern of clustering was found, significant local hot spots and cold spots were identified. The regions of Hiraan and Bay emerge as critical hot spots, where protracted conflict and climate shocks have decimated infrastructure and created a public health emergency. In stark contrast, the cold spots in the more stable and urbanized regions of Banadir and Lower Shabelle underscore the powerful role that governance, security, and humanitarian presence play in enabling access to services.

Ultimately, tackling Somalia's sanitation crisis demands a multi-pronged and evidence-informed approach. Interventions must be geographically targeted, prioritizing the identified hotspots with urgent investment in basic infrastructure. Policies must be tailored to the distinct needs of urban, rural, and nomadic populations, with a special focus on the highly vulnerable IDP communities by ensuring housing projects are always accompanied by essential water and sanitation services. Addressing these deep-rooted inequalities is not merely an infrastructure challenge; it is a critical imperative for advancing public health, upholding human dignity, especially for women and girls, and achieving sustainable development for all in Somalia.

## Author contributions

**Conceptualization:** Omar Muhumed Maidhane, Abdisalam Hassan Muse, Abdirahman Omer Osman, Nur Mohamud Ali, Shacban Abdilahi Elmi.

**Data curation:** Omar Muhumed Maidhane, Abdisalam Hassan Muse, Muse H Abdi, Nur Mohamud Ali.

**Formal analysis:** Omar Muhumed Maidhane, Abdirahman Omer Osman, Muse H Abdi, Nur Mohamud Ali, Shacban Abdilahi Elmi.

**Funding acquisition:** Omar Muhumed Maidhane.

**Investigation:** Omar Muhumed Maidhane, Abdirahman Omer Osman, Muse H Abdi, Shacban Abdilahi Elmi.

**Methodology:** Omar Muhumed Maidhane, Mahdi Hashi Hassan, Shacban Abdilahi Elmi.

**Project administration:** Omar Muhumed Maidhane.

**Resources:** Omar Muhumed Maidhane, Mahdi Hashi Hassan.

**Software:** Omar Muhumed Maidhane.

**Validation:** Omran Salih.

**Visualization:** Omar Muhumed Maidhane, Abdisalam Hassan Muse, Mahdi Hashi Hassan.

**Writing – original draft:** Omar Muhumed Maidhane.

**Writing – review & editing:** Omar Muhumed Maidhane.

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
