## [Decision Letter · Decision Letter 0]

19 Feb 2026

Dear Dr. Maidhane,

Thank you for submitting your manuscript to PLOS ONE. After careful consideration, we feel that it has merit but does not fully meet PLOS ONE’s publication criteria as it currently stands. Therefore, we invite you to submit a revised version of the manuscript that addresses the points raised during the review process.

We look forward to receiving your revised manuscript.

Kind regards,

Alfredo Luis Fort, M.D., M.Sc., Ph.D.

Academic Editor

PLOS One

**Journal Requirements:**

1. When submitting your revision, we need you to address these additional requirements. Please ensure that your manuscript meets PLOS ONE's style requirements, including those for file naming. The PLOS ONE style templates can be found at https://journals.plos.org/plosone/s/file?id=wjVg/PLOSOne_formatting_sample_main_body.pdf and https://journals.plos.org/plosone/s/file?id=ba62/PLOSOne_formatting_sample_title_authors_affiliations.pdf 2. Thank you for uploading your study's underlying data set. Unfortunately, the repository you have noted in your Data Availability statement does not qualify as an acceptable data repository according to PLOS's standards. At this time, please upload the minimal data set necessary to replicate your study's findings to a stable, public repository (such as figshare or Dryad) and provide us with the relevant URLs, DOIs, or accession numbers that may be used to access these data. For a list of recommended repositories and additional information on PLOS standards for data deposition, please see https://journals.plos.org/plosone/s/recommended-repositories. 3. We note that Figures 1, 3 and 4 in your submission contain map images which may be copyrighted. All PLOS content is published under the Creative Commons Attribution License (CC BY 4.0), which means that the manuscript, images, and Supporting Information files will be freely available online, and any third party is permitted to access, download, copy, distribute, and use these materials in any way, even commercially, with proper attribution. For these reasons, we cannot publish previously copyrighted maps or satellite images created using proprietary data, such as Google software (Google Maps, Street View, and Earth). For more information, see our copyright guidelines: http://journals.plos.org/plosone/s/licenses-and-copyright. We require you to either present written permission from the copyright holder to publish these figures specifically under the CC BY 4.0 license, or remove the figures from your submission: a. You may seek permission from the original copyright holder of Figures 1, 3 and 4 to publish the content specifically under the CC BY 4.0 license.   We recommend that you contact the original copyright holder with the Content Permission Form (http://journals.plos.org/plosone/s/file?id=7c09/content-permission-form.pdf) and the following text:“I request permission for the open-access journal PLOS ONE to publish XXX under the Creative Commons Attribution License (CCAL) CC BY 4.0 (http://creativecommons.org/licenses/by/4.0/). Please be aware that this license allows unrestricted use and distribution, even commercially, by third parties. Please reply and provide explicit written permission to publish XXX under a CC BY license and complete the attached form.” Please upload the completed Content Permission Form or other proof of granted permissions as an "Other" file with your submission. In the figure caption of the copyrighted figure, please include the following text: “Reprinted from [ref] under a CC BY license, with permission from [name of publisher], original copyright [original copyright year].” b. If you are unable to obtain permission from the original copyright holder to publish these figures under the CC BY 4.0 license or if the copyright holder’s requirements are incompatible with the CC BY 4.0 license, please either i) remove the figure or ii) supply a replacement figure that complies with the CC BY 4.0 license. Please check copyright information on all replacement figures and update the figure caption with source information. If applicable, please specify in the figure caption text when a figure is similar but not identical to the original image and is therefore for illustrative purposes only.The following resources for replacing copyrighted map figures may be helpful: USGS National Map Viewer (public domain): http://viewer.nationalmap.gov/viewer/The Gateway to Astronaut Photography of Earth (public domain): http://eol.jsc.nasa.gov/sseop/clickmap/Maps at the CIA (public domain): https://www.cia.gov/library/publications/the-world-factbook/index.html and https://www.cia.gov/library/publications/cia-maps-publications/index.htmlNASA Earth Observatory (public domain): http://earthobservatory.nasa.gov/Landsat:
http://landsat.visibleearth.nasa.gov/USGS EROS (Earth Resources Observatory and Science (EROS) Center) (public domain): http://eros.usgs.gov/#Natural Earth (public domain): http://www.naturalearthdata.com/ 4. Please upload a new copy of Figure 2, as the detail is not clear. Please follow the link for more information:  https://journals.plos.org/plosone/s/figures 5. If the reviewer comments include a recommendation to cite specific previously published works, please review and evaluate these publications to determine whether they are relevant and should be cited. There is no requirement to cite these works unless the editor has indicated otherwise. 

**Additional Editor Comments:**

Your manuscript has been read and comments are made by the reviewers. Please go through and modify/adapt as necessary. I have also included a file for you to see the suggested edits. Thank you.

Reviewers' comments:

**Comments to the Author**

1. Is the manuscript technically sound, and do the data support the conclusions?

Reviewer #1: Yes

Reviewer #2: Yes

2. Has the statistical analysis been performed appropriately and rigorously?

Reviewer #1: Yes

Reviewer #2: Yes

3. Have the authors made all data underlying the findings in their manuscript fully available?

Reviewer #1: Yes

Reviewer #2: Yes

4. Is the manuscript presented in an intelligible fashion and written in standard English?

Reviewer #1: Yes

Reviewer #2: Yes

**Reviewer #1:**  In general, the manuscript has some good information. English writing is acceptable. The manuscript can be accepted for publication after having some revisions as followings: In general, the manuscript has some good information. English writing is acceptable. The manuscript can be accepted for publication after having some revisions as followings:

- Abstract is too long. This should be shortened, in particular the background and method.

- Introduction: Please use some updated reference (within the past 10 years).

- Study area: Should provide brief information about population, population density, economic growth, etc.

- Section 2.2 and 2.3 should combine into one section to avoid repetition. Additionally, please provide brief information about the type of survey: interview or questionnaire forms, online or offline, etc.

- Section 3.1 needs to include:

o Discussion on the result for each variable. For instance, why the improved sanitation facilities were high in urban, low in rural; why it is different for marital status, etc.

o Comparison with previous studies in Somali or similar countries in Africa on the status of unimproved or improved sanitation.

- Section 3.2 needs to have a discussion on which Model is more reliable, why there were the difference/similarity among Model 1 and Model 3.

- Section 3.4.1, provide deeper discussion on why the improved sanitation facilities were high in Lower Shabelle or Banadir regions and low in Bay or Hiraan regions;

- Section 3.4.2 should compare the normal distribution in Somali with previous studies.

- The Discussion section: should state more about why there were such results. Some aspects as economic condition, level of public education, etc. should be analysed in details to support the results. After that, please propose some solutions to improve this issue.

- The conclusion should be rewritten with the information added from the discussion.

Conclusion: Major revision is required.

**Reviewer #2:**  1- The title is fine, but I think it's a little long. 1- The title is fine, but I think it's a little long.

I suggest this: Spatial distribution and multilevel determinants of unimproved sanitation in Somalia: evidence from SIHBS 2022

2- When reviewing your article, I noticed an inconsistency between the descriptive results and those of the multilevel model concerning the use of unimproved sanitation facilities. In the description, the prevalence among nomadic populations is 83.28%, while the corresponding OR is 0.04, which suggests the opposite. Similarly, for urban populations, the OR of 7.99 does not correspond to the 23.88% observed in the description.

Could you clarify or justify these discrepancies? It would be useful to understand whether this is the result of a particular coding of the dependent variable, a choice of reference category, or another model parameter. This clarification is essential for assessing the consistency and credibility of the results.

3- Justify why spatial + multilevel analysis adds scientific value.

Clarify what distinguishes this study from previous ones.

4- Open defecation is classified as unimproved. However, you then explain that low nomadic AOR is due to unrecorded open defecation. This contradicts your own definition. To be clarified.

5- The section on kriging lacks methodological rigor. You mention the use of ordinary kriging without presenting any cross-validation. Without indicators such as RMSE or Mean Error, predictive performance cannot be evaluated. No estimate of the standard error or prediction variance is provided, which makes it impossible to assess spatial uncertainty. The variogram is not described. You do not specify the model chosen or parameters such as nugget, sill, or range. This omission compromises reproducibility and weakens the scientific credibility of the spatial analysis.

6- Correct inconsistencies between “Result” and “Results.”

**Do you want your identity to be public for this peer review?** For information about this choice, including consent withdrawal, please see our For information about this choice, including consent withdrawal, please see our Privacy Policy .

Reviewer #1: **Yes:** Huyen Thi Thanh DangHuyen Thi Thanh Dang

Reviewer #2: **Yes:** Romaric Christian Marc HEKPAZORomaric Christian Marc HEKPAZO

---

## [Author Response · Author response to Decision Letter 1]

6 Apr 2026

Dear Dr. Alfredo Luis Fort,

Thank you for the opportunity to revise our manuscript, "Spatial Distribution and Determinants of Unimproved Sanitation Facilities Among Households in Somalia: Using Somalia Integrated Household Budget Survey (SIHBS 2022" (PONE-D-25-58376). We are grateful to you and the reviewers for the insightful and constructive feedback, which has significantly improved the quality of our paper

---

## [Editor Report · Decision Letter 1]

9 Apr 2026

Spatial Distribution and Determinants of Unimproved Sanitation Facilities Among Households in Somalia: Using Somalia Integrated Household Budget Survey (SIHBS 2022)

PONE-D-25-58376R1

Dear Dr. Maidhane,

We’re pleased to inform you that your manuscript has been judged scientifically suitable for publication and will be formally accepted for publication once it meets all outstanding technical requirements.

Kind regards,

Alfredo Luis Fort, M.D., M.Sc., Ph.D.

Academic Editor

PLOS One

Additional Editor Comments (optional):

You as authors have taken all comments from the reviewers and myself and added descriptions that were necessary to understand your very complex statistical analyses and results. However, there are still areas where some editing is necessary, but mostly as preparation for publication. Thus, I decided to move ahead with Acceptance and that such final edits are made as the manuscript is prepared for publication. Thanks.

---

## [Editor Report · Acceptance letter]

PONE-D-25-58376R1

PLOS One

Dear Dr. Maidhane,

I'm pleased to inform you that your manuscript has been deemed suitable for publication in PLOS One. Congratulations! Your manuscript is now being handed over to our production team.

Kind regards,

on behalf of

Dr. Alfredo Luis Fort

Academic Editor

PLOS One